# Unsupervised Transcription of Piano Music

**Taylor Berg-Kirkpatrick**    **Jacob Andreas**    **Dan Klein**
Computer Science Division
University of California, Berkeley
`{tberg,jda,klein}@cs.berkeley.edu`

## Abstract

We present a new probabilistic model for transcribing piano music from audio to a symbolic form. Our model reflects the process by which discrete musical events give rise to acoustic signals that are then superimposed to produce the observed data. As a result, the inference procedure for our model naturally resolves the source separation problem introduced by the the piano's polyphony. In order to adapt to the properties of a new instrument or acoustic environment being transcribed, we learn recording-specific spectral profiles and temporal envelopes in an unsupervised fashion. Our system outperforms the best published approaches on a standard piano transcription task, achieving a 10.6% relative gain in note onset $F_1$ on real piano audio.

## 1 Introduction

Automatic music transcription is the task of transcribing a musical audio signal into a symbolic representation (for example MIDI or sheet music). We focus on the task of transcribing piano music, which is potentially useful for a variety of applications ranging from information retrieval to musicology. This task is extremely difficult for multiple reasons. First, even individual piano notes are quite rich. A single note is not simply a fixed-duration sine wave at an appropriate frequency, but rather a full spectrum of harmonics that rises and falls in intensity. These profiles vary from piano to piano, and therefore must be learned in a recording-specific way. Second, piano music is generally *polyphonic*, i.e. multiple notes are played simultaneously. As a result, the harmonics of the individual notes can and do collide. In fact, combinations of notes that exhibit ambiguous harmonic collisions are particularly common in music, because consonances sound pleasing to listeners. This polyphony creates a source-separation problem at the heart of the transcription task.

In our approach, we learn the timbral properties of the piano being transcribed (i.e. the spectral and temporal shapes of each note) in an unsupervised fashion, directly from the input acoustic signal. We present a new probabilistic model that describes the process by which discrete musical events give rise to (separate) acoustic signals for each keyboard note, and the process by which these signals are superimposed to produce the observed data. Inference over the latent variables in the model yields transcriptions that satisfy an informative prior distribution on the discrete musical structure and at the same time resolve the source-separation problem.

For the problem of unsupervised piano transcription where the test instrument is not seen during training, the classic starting point is a non-negative factorization of the acoustic signal's spectrogram. Most previous work improves on this baseline in one of two ways: either by better modeling the discrete musical structure of the piece being transcribed [1, 2] or by better adapting to the timbral properties of the source instrument [3, 4]. Combining these two kinds of approaches has proven challenging. The standard approach to modeling discrete musical structures—using hidden Markov or semi-Markov models—relies on the availability of fast dynamic programs for inference. Here, coupling these discrete models with timbral adaptation and source separation breaks the conditional independence assumptions that the dynamic programs rely on. In order to avoid this inference problem, past approaches typically defer detailed modeling of discrete structure or timbre to a post-processing step [5, 6, 7].

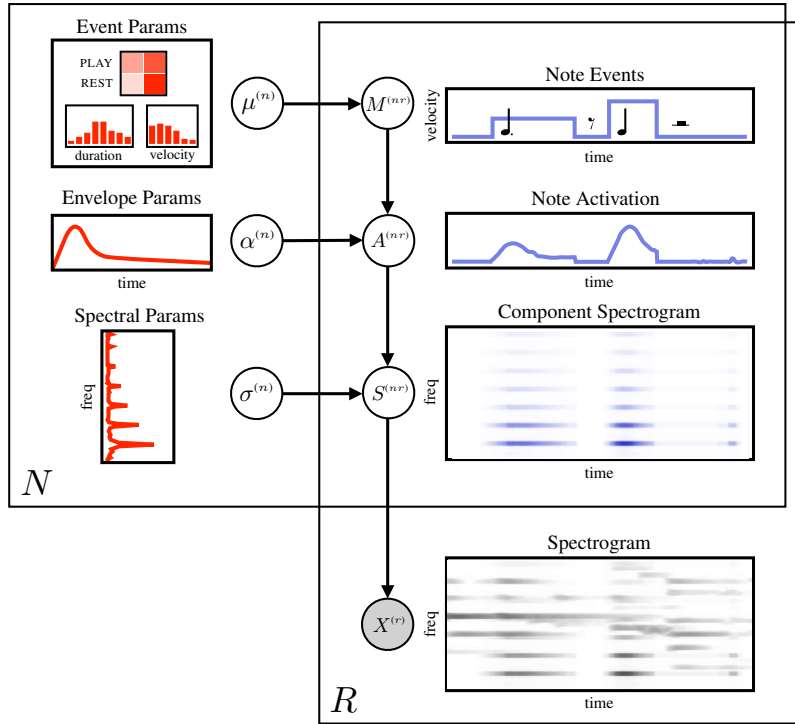

Figure 1: We transcribe a dataset consisting of $R$ songs produced by a single piano with $N$ notes. For each keyboard note, $n$, and each song, $r$, we generate a sequence of musical events, $M^{(nr)}$, parameterized by $\mu^{(n)}$. Then, conditioned on $M^{(nr)}$, we generate an activation time series, $A^{(nr)}$, parameterized by $\alpha^{(n)}$. Next, conditioned on $A^{(nr)}$, we generate a component spectrogram for note $n$ in song $r$, $S^{(nr)}$, parameterized by $\sigma^{(n)}$. The observed total spectrogram for song $r$ is produced by superimposing component spectrograms: $X^{(r)} = \sum_n S^{(nr)}$.

We present the first approach that tackles these discrete and timbral modeling problems jointly. We have two primary contributions: first, a new generative model that reflects the causal process underlying piano sound generation in an articulated way, starting with discrete musical structure; second, a tractable approximation to the inference problem over transcriptions and timbral parameters. Our approach achieves state-of-the-art results on the task of polyphonic piano music transcription. On a standard dataset consisting of real piano audio data, annotated with ground-truth onsets, our approach outperforms the best published models for this task on multiple metrics, achieving a 10.6% relative gain in our primary measure of note onset $F_1$.

## 2 Model

Our model is laid out in Figure 1. It has parallel random variables for each note on the piano keyboard. For now, we illustrate these variables for a single concrete note—say $C^\sharp$ in the 4th octave—and in Section 2.4 describe how the parallel components combine to produce the observed audio signal. Consider a single song, divided into $T$ time steps. The transcription will be $I$ musical events long. The component model for $C^\sharp$ consists of three primary random variables:

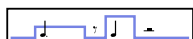
$M$, a sequence of $I$ symbolic musical events, analogous to the locations and values of symbols along the $C^\sharp$ staff line in sheet music,

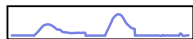
$A$, a time series of $T$ activations, analogous to the loudness of sound emitted by the $C^\sharp$ piano string over time as it peaks and attenuates during each event in $M$,

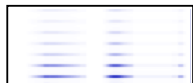
$S$, a spectrogram of $T$ frames, specifying the spectrum of frequencies over time in the acoustic signal produced by the $C^\sharp$ string.

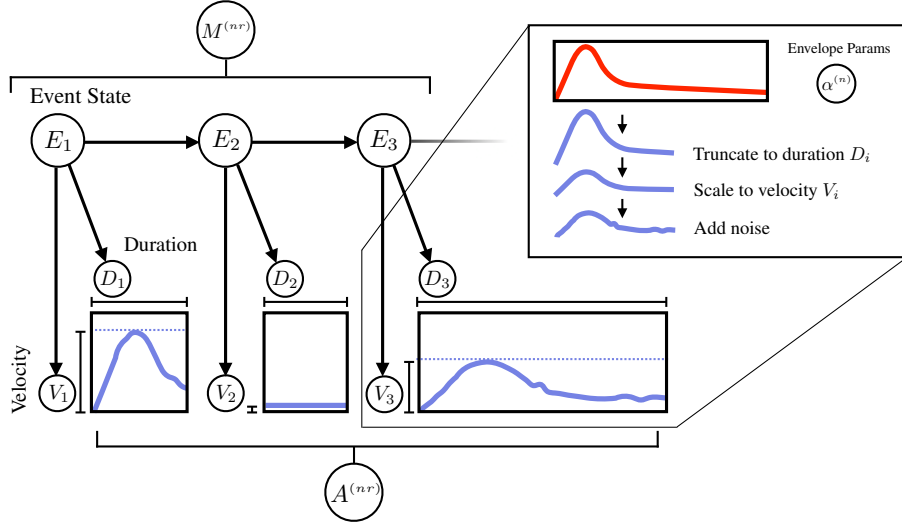

Figure 2: Joint distribution on musical events, $M^{(nr)}$, and activations, $A^{(nr)}$, for note $n$ in song $r$, conditioned on event parameters, $\mu^{(n)}$, and envelope parameters, $\alpha^{(n)}$. The dependence of $E_i$, $D_i$, and $V_i$ on $n$ and $r$ is suppressed for simplicity.

The parameters that generate each of these random variables are depicted in Figure 1. First, musical events, $M$, are generated from a distribution parameterized by $\mu^{(C^\sharp)}$, which specifies the probability that the $C^\sharp$ key is played, how long it is likely to be held for (duration), and how hard it is likely to be pressed (velocity). Next, the activation of the $C^\sharp$ string over time, $A$, is generated conditioned on $M$ from a distribution parameterized by a vector, $\alpha^{(C^\sharp)}$ (see Figure 1), which specifies the shape of the rise and fall of the string's activation each time the note is played. Finally, the spectrogram, $S$, is generated conditioned on $A$ from a distribution parameterized by a vector, $\sigma^{(C^\sharp)}$ (see Figure 1), which specifies the frequency distribution of sounds produced by the $C^\sharp$ string. As depicted in Figure 3, $S$ is produced from the outer product of $\sigma^{(C^\sharp)}$ and $A$. The joint distribution for the note[1] is:

$$
\begin{aligned}
P(S, A, M | \sigma^{(C^\sharp)}, \alpha^{(C^\sharp)}, \mu^{(C^\sharp)}) \;=\; & P(M|\mu^{(C^\sharp)}) && \text{[Event Model, Section 2.1]} \\
& \cdot\, P(A|M, \alpha^{(C^\sharp)}) && \text{[Activation Model, Section 2.2]} \\
& \cdot\, P(S|A, \sigma^{(C^\sharp)}) && \text{[Spectrogram Model, Section 2.3]}
\end{aligned}
$$

In the next three sections we give detailed descriptions for each of the component distributions.

## 2.1 Event Model

Our symbolic representation of musical structure (see Figure 2) is similar to the MIDI format used by musical synthesizers. $M$ consists of a sequence of $I$ random variables representing musical events for the $C^\sharp$ piano key: $M = (M_1, M_2, \ldots, M_I)$. Each event $M_i$, is a tuple consisting of a state, $E_i$, which is either PLAY or REST, a duration $D_i$, which is a length in time steps, and a velocity $V_i$, which specifies how hard the key was pressed (assuming $E_i$ is PLAY).

The graphical model for the process that generates $M$ is depicted in Figure 2. The sequence of states, $(E_1, E_2, \ldots, E_I)$, is generated from a Markov model. The transition probabilities, $\mu^{\text{TRANS}}$, control how frequently the note is played (some notes are more frequent than others). An event's duration, $D_i$, is generated conditioned on $E_i$ from a distribution parameterized by $\mu^{\text{DUR}}$. The durations of PLAY events have a multinomial parameterization, while the durations of REST events are distributed geometrically. An event's velocity, $V_i$, is a real number on the unit interval and is generated conditioned on $E_i$ from a distribution parameterized by $\mu^{\text{VEL}}$. If $E_i = $ REST, deterministically $V_i = 0$. The complete event parameters for keyboard note $C^\sharp$ are $\mu^{(C^\sharp)} = (\mu^{\text{TRANS}}, \mu^{\text{DUR}}, \mu^{\text{VEL}})$.

## 2.2 Activation Model

In an actual piano, when a key is pressed, a hammer strikes a string and a sound with sharply rising volume is emitted. The string continues to emit sound as long as the key remains depressed, but the volume decays since no new energy is being transferred. When the key is released, a damper falls back onto the string, truncating the decay. Examples of this trajectory are depicted in Figure 1 in the graph of activation values. The graph depicts the note being played softly and held, and then being played more loudly, but held for a shorter time. In our model, PLAY events represent hammer strikes on a piano string with raised damper, while REST events represent the lowered damper. In our parameterization, the shape of the rise and decay is shared by all PLAY events for a given note, regardless of their duration and velocity. We call this shape an *envelope* and describe it using a positive vector of parameters. For our running example of $C^\sharp$, this parameter vector is $\alpha^{(C^\sharp)}$ (depicted to the right). 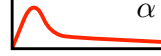

The time series of activations for the $C^\sharp$ string, $A$, is a positive vector of length $T$, where $T$ denotes the total length of the song in time steps. Let $[A]_t$ be the activation at time step $t$. As mentioned in Section 2, $A$ may be thought of as roughly representing the loudness of sound emitted by the piano string as a function of time. The process that generates $A$ is depicted in Figure 2. We generate $A$ conditioned on the musical events, $M$. Each musical event, $M_i = (E_i, D_i, V_i)$, produces a segment of activations, $A_i$, of length $D_i$. For PLAY events, $A_i$ will exhibit an increase in activation. For REST events, the activation will remain low. The segments are appended together to make $A$. The activation values in each segment are generated in a way loosely inspired by piano mechanics. Given $\alpha^{(C^\sharp)}$, we generate the values in segment $A_i$ as follows: $\alpha^{(C^\sharp)}$ is first truncated to duration $D_i$, then is scaled by the velocity of the strike, $V_i$, and, finally, is used to parameterize an activation noise distribution which generates the segment $A_i$. Specifically, we add independent Gaussian noise to each dimension after $\alpha^{(C^\sharp)}$ is truncated and scaled. In principle, this choice of noise distribution gives a formally deficient model, since activations are positive, but in practice performs well and has the benefit of making inference mathematically simple (see Section 3.1).

## 2.3 Component Spectrogram Model

Piano sounds have a harmonic structure; when played, each piano string primarily emits energy at a fundamental frequency determined by the string's length, but also at all integer multiples of that frequency (called partials) with diminishing strength (see the depiction to the right). For example, the audio signal produced by the $C^\sharp$ string will vary in loudness, but its frequency distribution will remain mostly fixed. We call this frequency distribution a *spectral profile*. In our parameterization, the spectral profile of $C^\sharp$ is specified by a positive spectral parameter vector, $\sigma^{(C^\sharp)}$ (depicted to the right). $\sigma^{(C^\sharp)}$ is a vector of length $F$, where $[\sigma^{(C^\sharp)}]_f$ represents the weight of frequency bin $f$. 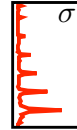

In our model, the audio signal produced by the $C^\sharp$ string over the course of the song is represented as a spectrogram, $S$, which is a positive matrix with $F$ rows, one for each frequency bin, $f$, and $T$ columns, one for each time step, $t$ (see Figures 1 and 3 for examples). We denote the magnitude of frequency $f$ at time step $t$ as $[S]_{ft}$. In order to generate the spectrogram (see Figure 3), we first produce a matrix of intermediate values by taking the outer product of the spectral profile, $\sigma^{(C^\sharp)}$, and the activations, $A$. These intermediate values are used to parameterize a spectrogram noise distribution that generates $S$. Specifically, for each frequency bin $f$ and each time step $t$, the corresponding value of the spectrogram, $[S]_{ft}$, is generated from a noise distribution parameterized by $[\sigma^{(C^\sharp)}]_f \cdot [A]_t$. In practice, the choice of noise distribution is very important. After examining residuals resulting from fitting mean parameters to actual musical spectrograms, we experimented with various noising assumptions, including multiplicative gamma noise, additive Gaussian noise, log-normal noise, and Poisson noise. Poisson noise performed best. This is consistent with previous findings in the literature, where non-negative matrix factorization using KL divergence (which has a generative interpretation as maximum likelihood inference in a Poisson model [8]) is commonly chosen [7, 2]. Under the Poisson noise assumption, the spectrogram is interpreted as a matrix of (large) integer counts.

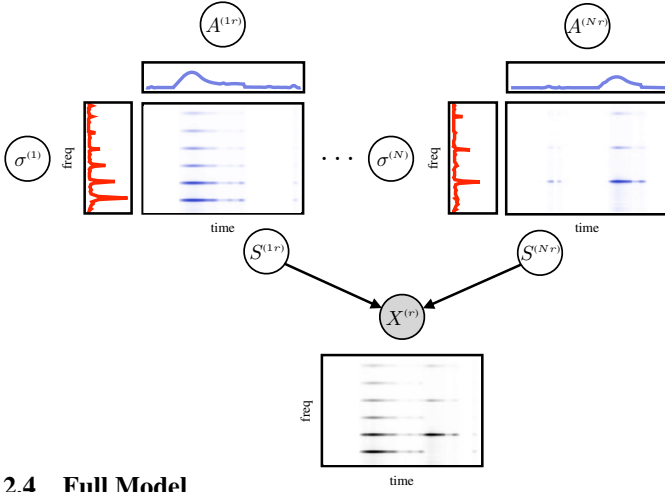

Figure 3: Conditional distribution for song $r$ on the observed total spectrogram, $X^{(r)}$, and the component spectrograms for each note, $(S^{(1r)}, \ldots, S^{(Nr)})$, given the activations for each note, $(A^{(1r)}, \ldots, A^{(Nr)})$, and spectral parameters for each note, $(\sigma^{(1)}, \ldots, \sigma^{(N)})$. $X^{(r)}$ is the superposition of the component spectrograms: $X^{(r)} = \sum_n S^{(nr)}$.

## 2.4 Full Model

So far we have only looked at the component of the model corresponding to a single note's contribution to a single song. Our full model describes the generation of a collection of many songs, from a complete instrument with many notes. This full model is diagrammed in Figures 1 and 3. Let a piano keyboard consist of $N$ notes (on a standard piano, $N$ is 88), where $n$ indexes the particular note. Each note, $n$, has its own set of musical event parameters, $\mu^{(n)}$, envelope parameters, $\alpha^{(n)}$, and spectral parameters, $\sigma^{(n)}$. Our corpus consists of $R$ songs ("recordings"), where $r$ indexes a particular song. Each pair of note $n$ and song $r$ has it's own musical events variable, $M^{(nr)}$, activations variable, $A^{(nr)}$, and component spectrogram $S^{(nr)}$. The full spectrogram for song $r$, which is the observed input, is denoted as $X^{(r)}$. Our model generates $X^{(r)}$ by superimposing the component spectrograms: $X^{(r)} = \sum_n S^{(nr)}$. Going forward, we will need notation to group together variables across all $N$ notes: let $\mu = (\mu^{(1)}, \ldots, \mu^{(N)})$, $\alpha = (\alpha^{(1)}, \ldots, \alpha^{(N)})$, and $\sigma = (\sigma^{(1)}, \ldots, \sigma^{(N)})$. Also let $M^{(r)} = (M^{(1r)}, \ldots, M^{(Nr)})$, $A^{(r)} = (A^{(1r)}, \ldots, A^{(Nr)})$, and $S^{(r)} = (S^{(1r)}, \ldots, S^{(Nr)})$.

## 3 Learning and Inference

Our goal is to estimate the unobserved musical events for each song, $M^{(r)}$, as well as the unknown envelope and spectral parameters of the piano that generated the data, $\alpha$ and $\sigma$. Our inference will estimate both, though our output is only the musical events, which specify the final transcription. Because MIDI sample banks (piano synthesizers) are readily available, we are able to provide the system with samples from generic pianos (but not from the piano being transcribed). We also give the model information about the distribution of notes in real musical scores by providing it with an external corpus of symbolic music data. Specifically, the following information is available to the model during inference: 1) the total spectrogram for each song, $X^{(r)}$, which is the input, 2) the event parameters, $\mu$, which we estimate by collecting statistics on note occurrences in the external corpus of symbolic music, and 3) truncated normal priors on the envelopes and spectral profiles, $\alpha$ and $\sigma$, which we extract from the MIDI samples.

Let $\bar{M} = (M^{(1)}, \ldots, M^{(R)})$, $\bar{A} = (A^{(1)}, \ldots, A^{(R)})$, and $\bar{S} = (S^{(1)}, \ldots, S^{(R)})$, the tuples of event, activation, and spectrogram variables across all notes $n$ and songs $r$. We would like to compute the posterior distribution on $\bar{M}$, $\alpha$, and $\sigma$. However, marginalizing over the activations $\bar{A}$ couples the discrete musical structure with the superposition process of the component spectrograms in an intractable way. We instead approximate the joint MAP estimates of $\bar{M}$, $\bar{A}$, $\alpha$, and $\sigma$ via iterated conditional modes [9], only marginalizing over the component spectrograms, $\bar{S}$. Specifically, we perform the following optimization via block-coordinate ascent:

$$\max_{\bar{M}, \bar{A}, \alpha, \sigma} \prod_r \left[ \sum_{S^{(r)}} P(X^{(r)}, S^{(r)}, A^{(r)}, M^{(r)} | \mu, \alpha, \sigma) \right] \cdot P(\alpha, \sigma)$$

The updates for each group of variables are described in the following sections: $\bar{M}$ in Section 3.1, $\alpha$ in Section 3.2, $\bar{A}$ in Section 3.3, and $\sigma$ in Section 3.4.

## 3.1 Updating Events

We update $\bar{M}$ to maximize the objective while the other variables are held fixed. The joint distribution on $\bar{M}$ and $\bar{A}$ is a hidden semi-Markov model [10]. Given the optimal velocity for each segment of activations, the computation of the emission potentials for the semi-Markov dynamic program is straightforward and the update over $\bar{M}$ can be performed exactly and efficiently. We let the distribution of velocities for PLAY events be uniform. This choice, together with the choice of Gaussian activation noise, yields a simple closed-form solution for the optimal setting of the velocity variable $V_i^{(nr)}$. Let $[\alpha^{(n)}]_j$ denote the $j$th value of the envelope vector $\alpha^{(n)}$. Let $[A_i^{(nr)}]_j$ be the $j$th entry of the segment of $A^{(nr)}$ generated by event $M_i^{(nr)}$. The velocity that maximizes the activation segment's probability is given by:

$$V_i^{(nr)} = \frac{\sum_{j=1}^{D_i^{(nr)}} \left( [\alpha^{(n)}]_j \cdot [A_i^{(nr)}]_j \right)}{\sum_{j=1}^{D_i^{(nr)}} [\alpha^{(n)}]_j^2}$$

## 3.2 Updating Envelope Parameters

Given settings of the other variables, we update the envelope parameters, $\alpha$, to optimize the objective. The truncated normal prior on $\alpha$ admits a closed-form update. Let $I(j, n, r) = \{i : D_i^{(nr)} \le j\}$, the set of event indices for note $n$ in song $r$ with durations no longer than $j$ time steps. Let $[\alpha_0^{(n)}]_j$ be the location parameter for the prior on $[\alpha^{(n)}]_j$, and let $\beta$ be the scale parameter (which is shared across all $n$ and $j$). The update for $[\alpha^{(n)}]_j$ is given by:

$$[\alpha^{(n)}]_j = \frac{\sum_{(n,r)} \sum_{i \in I(j,n,r)} V_i^{(nr)} [A_i^{(nr)}]_j + \frac{1}{2\beta^2} [\alpha_0^{(n)}]_j}{\sum_{(n,r)} \sum_{i \in I(j,n,r)} [A_i^{(nr)}]_j^2 + \frac{1}{2\beta^2}}$$

## 3.3 Updating Activations

In order to update the activations, $\bar{A}$, we optimize the objective with respect to $\bar{A}$, with the other variables held fixed. The choice of Poisson noise for generating each of the component spectrograms, $S^{(nr)}$, means that the conditional distribution of the total spectrogram for each song, $X^{(r)} = \sum_n S^{(nr)}$, with $S^{(r)}$ marginalized out, is also Poisson. Specifically, the distribution of $[X^{(r)}]_{ft}$ is Poisson with mean $\sum_n \left( [\sigma^{(n)}]_f \cdot [A^{(nr)}]_t \right)$. Optimizing the probability of $X^{(r)}$ under this conditional distribution with respect to $A^{(r)}$ corresponds to computing the supervised NMF using KL divergence [7]. However, to perform the correct update in our model, we must also incorporate the distribution of $A^{(r)}$, and so, instead of using the standard multiplicative NMF updates, we use exponentiated gradient ascent [11] on the log objective. Let $L$ denote the log objective, let $\tilde{\alpha}(n, r, t)$ denote the velocity-scaled envelope value used to generate the activation value $[A^{(nr)}]_t$, and let $\gamma^2$ denote the variance parameter for the Gaussian activation noise. The gradient of the log objective with respect to $[A^{(nr)}]_t$ is:

$$\frac{\partial L}{\partial [A^{(nr)}]_t} = \sum_f \left[ \frac{[X^{(r)}]_{ft} \cdot [\sigma^{(n)}]_f}{\sum_{n'} \left( [\sigma^{(n')}]_f \cdot [A^{(n'r)}]_t \right)} - [\sigma^{(n)}]_f \right] - \frac{1}{\gamma^2} \left( [A^{(nr)}]_t - \tilde{\alpha}(n, r, t) \right)$$

## 3.4 Updating Spectral Parameters

The update for the spectral parameters, $\sigma$, is similar to that of the activations. Like the activations, $\sigma$ is part of the parameterization of the Poisson distribution on each $X^{(r)}$. We again use exponentiated gradient ascent. Let $[\sigma_0^{(n)}]_f$ be the location parameter of the prior on $[\sigma^{(n)}]_f$, and let $\xi$ be the scale parameter (which is shared across all $n$ and $f$). The gradient of the the log objective with respect to $[\sigma^{(n)}]_f$ is given by:

$$\frac{\partial L}{\partial [\sigma^{(n)}]_f} = \sum_{(r,t)} \left[ \frac{[X^{(r)}]_{ft} \cdot [A^{(nr)}]_t}{\sum_{n'} \left( [\sigma^{(n')}]_f \cdot [A^{(n'r)}]_t \right)} - [A^{(nr)}]_t \right] - \frac{1}{\xi^2} \left( [\sigma^{(n)}]_f - [\sigma_0^{(n)}]_f \right)$$

# 4 Experiments

Because polyphonic transcription is so challenging, much of the existing literature has either worked with synthetic data [12] or assumed access to the test instrument during training [5, 6, 13, 7]. As our ultimate goal is the transcription of arbitrary recordings from real, previously-unseen pianos, we evaluate in an unsupervised setting, on recordings from an acoustic piano not observed in training.

**Data**    We evaluate on the MIDI-Aligned Piano Sounds (MAPS) corpus [14]. This corpus includes a collection of piano recordings from a variety of time periods and styles, performed by a human player on an acoustic "Disklavier" piano equipped with electromechanical sensors under the keys. The sensors make it possible to transcribe directly into MIDI while the instrument is in use, providing a ground-truth transcript to accompany the audio for the purpose of evaluation. In keeping with much of the existing music transcription literature, we use the first 30 seconds of each of the 30 ENSTDkAm recordings as a development set, and the first 30 seconds of each of the 30 ENSTDkCl recordings as a test set. We also assume access to a collection of synthesized piano sounds for parameter initialization, which we take from the MIDI portion of the MAPS corpus, and a large collection of symbolic music data from the IMSLP library [15, 16], used to estimate the event parameters in our model.

**Preprocessing**    We represent the input audio as a magnitude spectrum short-time Fourier transform with a 4096-frame window and a hop size of 512 frames, similar to the approach used by Weninger et al. [7]. We temporally downsample the resulting spectrogram by a factor of 2, taking the maximum magnitude over collapsed bins. The input audio is recorded at 44.1 kHz and the resulting spectrogram has 23ms frames.

**Initialization and Learning**    We estimate initializers and priors for the spectral parameters, $\sigma$, and envelope parameters, $\alpha$, by fitting isolated, synthesized, piano sounds. We collect these isolated sounds from the MIDI portion of MAPS, and average the parameter values across several synthesized pianos. We estimate the event parameters $\mu$ by counting note occurrences in the IMSLP data. At decode time, to fit the spectral and envelope parameters and predict transcriptions, we run 5 iterations of the block-coordinate ascent procedure described in Section 3.

**Evaluation**    We report two standard measures of performance: an *onset* evaluation, in which a predicted note is considered correct if it falls within 50ms of a note in the true transcription, and a *frame-level* evaluation, in which each transcription is converted to a boolean matrix specifying which notes are active at each time step, discretized to 10ms frames. Each entry is compared to the corresponding entry in the true matrix. Frame-level evaluation is sensitive to offsets as well as onsets, but does not capture the fact that note onsets have greater musical significance than do offsets. As is standard, we report precision (P), recall (R), and $F_1$-measure ($F_1$) for each of these metrics.

## 4.1 Comparisons

We compare our system to three state-of-the-art unsupervised systems: the hidden semi-Markov model described by Benetos and Weyde [2] and the spectrally-constrained factorization models described by Vincent et al. [3] and O'Hanlon and Plumbley [4]. To our knowledge, Benetos and Weyde [2] report the best published onset results for this dataset, and O'Hanlon and Plumbley [4] report the best frame-level results.

The literature also includes a number of supervised approaches to this task. In these approaches, a model is trained on annotated recordings from a known instrument. While best performance is achieved when testing on the same instrument used for training, these models can also achieve reasonable performance when applied to new instruments. Thus, we also compare to a discriminative baseline, a simplified reimplementation of a state-of-the-art supervised approach [7] which achieves slightly better performance than the original on this task. This system only produces note onsets, and therefore is not evaluated at a frame-level. We train the discriminative baseline on synthesized audio with ground-truth MIDI annotations, and apply it directly to our test instrument, which the system has never seen before.

| System | Onsets | | | Frames | | |
|---|---|---|---|---|---|---|
| | P | R | $F_1$ | P | R | $F_1$ |
| Discriminative [7] | 76.8 | 65.1 | 70.4 | - | - | - |
| Benetos [2] | - | - | 68.6 | - | - | 68.0 |
| Vincent [3] | 62.7 | 76.8 | 69.0 | 79.6 | 63.6 | 70.7 |
| O'Hanlon [4] | 48.6 | 73.0 | 58.3 | 73.4 | 72.8 | 73.2 |
| This work | 78.1 | 74.7 | **76.4** | 69.1 | 80.7 | **74.4** |

Table 1: Unsupervised transcription results on the MAPS corpus. "Onsets" columns show scores for identification (within $\pm 50$ms) of note start times. "Frames" columns show scores for 10ms frame-level evaluation. Our system achieves state-of-the-art results on both metrics.[2]

## 4.2 Results

Our model achieves the best published numbers on this task: as shown in Table 1, it achieves an onset $F_1$ of 76.4, which corresponds to a 10.6% relative gain over the onset $F_1$ achieved by the system of Vincent et al. [3], the top-performing unsupervised baseline on this metric. Surprisingly, the discriminative baseline [7], which was not developed for the unsupervised task, outperforms all the unsupervised baselines in terms of onset evaluation, achieving an $F_1$ of 70.4. Evaluated on frames, our system achieves an $F_1$ of 74.4, corresponding to a more modest 1.6% relative gain over the system of O'Hanlon and Plumbley [4], which is the best performing baseline on this metric.

The surprisingly competitive discriminative baseline shows that it is possible to achieve high onset accuracy on this task without adapting to the test instrument. Thus, it is reasonable to ask how much of the gain our model achieves is due to its ability to learn instrument timbre. If we skip the block-coordinate ascent updates (Section 3) for the envelope and spectral parameters, and thus prevent our system from adapting to the test instrument, onset $F_1$ drops from 76.4 to 72.6. This result indicates that learning instrument timbre does indeed help performance.

As a short example of our system's behavior, Figure 4 shows our system's output passed through a commercially-available MIDI-to-sheet-music converter. This example was chosen because its onset $F_1$ of 75.5 and error types are broadly representative of the system's performance on our data. The resulting score has musically plausible errors.

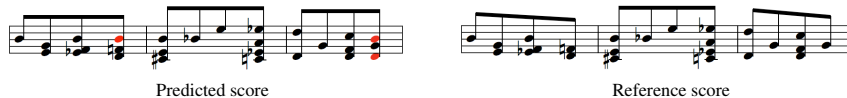

<div align="center">Predicted score          Reference score</div>

Figure 4: Result of passing our system's prediction and the reference transcription MIDI through the Garage-Band MIDI-to-sheet-music converter. This is a transcription of the first three bars of Schumann's *Hobgoblin*.

A careful inspection of the system's output suggests that a large fraction of errors are either off by an octave (i.e. the frequency of the predicted note is half or double the correct frequency) or are segmentation errors (in which a single key press is transcribed as several consecutive key presses). While these are tricky errors to correct, they may also be relatively harmless for some applications because they are not detrimental to musical perception: converting the transcriptions back to audio using a synthesizer yields music that is qualitatively quite similar to the original recordings.

## 5 Conclusion

We have shown that combining unsupervised timbral adaptation with a detailed model of the generative relationship between piano sounds and their transcriptions can yield state-of-the-art performance. We hope that these results will motivate further joint approaches to unsupervised music transcription. Paths forward include exploring more nuanced timbral parameterizations and developing more sophisticated models of discrete musical structure.

## Footnotes

[1]For notational convenience, we suppress the $C^\sharp$ superscripts on $M$, $A$, and $S$ until Section 2.4.

[2] For consistency we re-ran all systems in this table with our own evaluation code (except for the system of Benetos and Weyde [2], for which numbers are taken from the paper). For O'Hanlon and Plumbley [4] scores are higher than the authors themselves report; this is due to an extra post-processing step suggested by O'Hanlon in personal correspondence.

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
