[Reviews · NeurIPS 2014]

Submitted by Assigned_Reviewer_8

Quality and originality: The main claims of the authors make intuitive sense. Specifically, figure 1 presents a generative model which separates note onsets from activation and spectral information. This is in keeping with the physics of a piano, where a pianist initiates a note onset by sending the hammer in free-flight. This activates the string, generating harmonics. Those harmonics change over time based both on decay and on the piano’s physical dampers. The model is divided, thusly, into an Event Model, Activation Model, and Component Spectrogram Model with clear sections both on model details and inference details provided. The paper is of high quality in terms of presentation, linkage to actual instrument physics and experimental details. It’s original in terms of the specific factorizations applied, though in fact the main technical contribution seems to be that the method incorporates both discrete and timbral factors into the same model.

Clarity and significance: The paper is well-written and is easy to read. I question the significance of the model for a NIPS crowd. The underlying machine learning is pretty simple. I found it interesting for the way the authors build a model well suited to the acoustic piano. The results represent a good advancement on the state-of-the-art for a relevant MIR task, but it may be that this paper isn’t significant enough to warrant a NIPS paper. Maybe it’s a better ISMIR (music information retrieval conference) paper than a NIPS paper. I’m on the fence and would be happy to see it published here.
Summary: A probabilistic model for piano transcription is proposed and evaluated. The main advance lies in how the model factorizes note events, note activation and spectral components in a way that resolves the hard problem of source separation for piano. The model performs well on standard benchmarks.

Submitted by Assigned_Reviewer_15

The paper focuses on the important problem of polyphonic transcription in music information analysis/retrieval. The paper suggest a novel generative model with inspiration form piano music; however, the model might be generalized to other instruments, or maybe multiple instrument tracks. An (unsupervised) inference algorithm is developed, and the method is compared to state-of-the-art methods on the MAPS corpus data. Superior performance is demonstrated both in onset and frame-level evaluation metrics. Despite issues mentioned in detailed comments below, the paper is a solid and significant contribution that might be of interest to the NIPS community.

Quality:
The paper is relevant and well-written and describe in good detail a novel generative piano music model and arguments for most of the involved model ingredients (some issues mentioned in detailed comments). The model is sufficiently linked and contrasted to existing state-of-art techniques. The methods benefits from being unsupervised and has some generalizability to new instruments although this has only been tested in a rather limited setting. The performance as regards onset and frame-level metrics is superior to state-of-art techniques and the error patterns is argued to bee musically plausible despite the fact that such issues have not been directly been addressed in the model. Whether this is a unique feature of the proposed framework (over alternative methods) is, however, not discussed. The overall conclusion is that the paper is a significant contribution to the community and might inspire further development. Issues and needed clarifications are addressed in the detailed comments below.

Clarity:
• The paper is well-written and clearly formulated except issues addressed below. It is not possible to do exact reproduction of the results without more details - in particular as regards the experiments.
• The references are relevant and adequate.

Significance:
• The technical contribution seems sufficiently interesting to a part of the NIPS community related to audio/music, and to sequential models in general. The modeling framework is not complete as the adaption to novel un-seen instruments is not explicitly addressed (except implicit general nature noise models). Further, musical constraints which might help producing perceptually acceptable errors is not addressed. Finally only a single but feasible inference scheme is suggested, hence, the potential benefits of more advanced inference methods is not elicited.

Originality:
• The overall modeling framework is original and tries to include more aspects of the generation of the oberserved music spectrum.

Detailed comments:
• Line 033: test data > training data
• Last part of intro lines 087-091 should go to the conclusion section.
• Line 104 (and other places): loudness is a subjective/perceptual quality. Better to use amplitude/ intensity / strength.
• Line 181: activation will remain low? Do you mean low or zero?
• Line 185: Gaussian noise distribution. What is this stochastic variable going to model? How do you handle potential negative activations?
• Sec 2.3: What is the noise in the spectrum going to represent?
• Sec 2.3: Do you have any evidence for the claim that Poisson is the best noise model? Maybe refer to an available technical report.
• How is the priors on alpha and sigma going to be constructed, and how important/sensitive is the method to such choices? Have you considered music genre/style variations?
• Iterated conditional mode is the chosen as theinference method. Have you considered alternatives, and what are the arguments for sufficiency in the present case.
• Sec 3.1. Some details on inferring M is missing. Provide a complete description.
• How are the hyper-parameters beta, gamma and xi specified?
• The experimental section is rather brief. For reproducibility reasons it would be relevant to refer to a technical report on the details.
• The difference between ENSTDkAm and ENSTDkC1 is ambient or close-in recording of the Yamaha Diskklavier. Hence they might be rather similar! Please make clear that the generalizability to a bigger variety of pianos is unclear.
• Why is 23ms (actually 23,22ms) a good window size?
• Why is 50ms used in evaluation of onsets. 50ms corresponds to 1200 bpm and notes played above this rate might perceptually sound as a single tone. That could be part of the explanation.
• Explain in more detail the music possible errors in line 402. Do the alternative methods have same desirable property?
• Lines 418-420: Observed errors are claimed not to affect the re-synthesized audio. What is the evidence? How could you take such tolerable error patterns into consideration in the modeling framework?
• The generalizability of the model to other instruments or maybe polyphonic music as such is not addressed. Please include some discussion.
• Some references seem to be incomplete and ordering of text e.g. 'In Proceedings' should be reversed.
Summary: The paper introduces a novel generative model for polyphonic (piano) music. The model is presents a simple unsupervised inference from observed spectrograms of piano music. Superior performance in contrast to state-of-art methods has been reported. The conclusion of adapting to timbre of unseen pianos is somewhat limited due to experimental setup.

Submitted by Assigned_Reviewer_41

This is an interesting paper taking NIPS-style theoretical research to a practical problem that has been studied by a large community not typically attending NIPS. The authors are clearly aware of the work that has been done elsewhere on this topic, and significantly add to that to arrive at a superior system for polyphonic piano transcription.
The empirical evaluation is strong and well conducted.

Two very minor issues:
- "the the" in the abstract
- in section 2.1, I do not like the choice of 'transition probabilities' to refer to these \mu parameters, as it suggests they have a similar function as transition probabilities in HMMs (which, I understand, is not the case).
- I am wondering if a constant-Q transform would not yield a better performance. Is the frequency resolution of the spectrogram in the low frequency range sufficient to discriminate between different notes there? A discussion of this issue would be useful, e.g. showing the distribution of errors made across the frequency range.
Summary: This is a solid application paper.
The application is important and well-described.
The proposed solution strategy is adequate, non-trivial and rigorously described.
Author Feedback
Author rebuttal: We thank the reviewers for their helpful comments! We’ve replied to a range of reviewer points below. Specific suggestions about organiziation and copyediting will be incorporated in future revisions.

POISSON NOISE MODEL: We tried several different spectrogram noise models in our own experiments and found that the Poisson model gave the best transcription results. Other authors have performed similar analyses. Most notably, Peeling et al. (2010, "Generative spectrogram factorization models for polyphonic piano transcription") ran a similar evaluation and came to a similar conclusion.

INFERENCE FOR EVENT VARIABLES (i.e. M): We agree that an expanded description of the inference procedure for the musical events would add clarity and we’ll see what expansion can be done in the space available (additional details could also appear in a technical report).

HYPER-PARAMETER SELECTION: We used a held-out data to fit the parameters of the priors (beta, gamma, and xi). We set them (manually) to maximize transcription accuracy. We found that performance was not very sensitive to the settings of these hyper-parameters. We'll be sure to include this information in future revisions.

GENERALIZATION ACROSS PIANOS: While the ENSTDkAm and ENSTDkCl datasets have different acoustic properties (to the point where supervised methods trained on one set perform quite poorly on the other), it is definitely true that both use the Yamaha Disklavier (we do not know of data in this form from other sources, though we are interested in collecting it). We'll make this more clear in future revisions.

CONSTANT-Q TRANSFORM: The reviewer's suggestion about using the constant-Q transform is a good one. We had considered this approach, and originally opted against it at after finding results in the literature where a direct comparison of constant-Q and STFT came out against constant-Q (Weninger et. al, 2013, "A discriminative approach to polyphonic piano note transcription using supervised non-negative matrix factorization"). Our own analyses of system performance did not show substantially higher error rates in the lower registers. If space allows, we will present this analysis.

REPRODUCIBILITY: The best way we know to ensure reproducibility is to release all of our learning and evaluation code, which we plan to do.